# Analysis of Factors Affecting the High Subjective Well-Being of Chinese Residents Based on the 2014 China Family Panel Study

**DOI:** 10.3390/ijerph16142566

**Published:** 2019-07-18

**Authors:** Wen Xu, Haiyan Sun, Bo Zhu, Wei Bai, Xiao Yu, Ruixin Duan, Changgui Kou, Wenjun Li

**Affiliations:** 1Department of Social Medicine and Health Management, School of Public Health, Jilin University, Changchun 130021, China; 2Department of Epidemiology and Biostatistics, School of Public Health, Jilin University, Changchun 130021, China

**Keywords:** subjective well-being (SWB), psychological well-being (PWB), education, social trust, social relationship, physical exercise

## Abstract

(1) Purpose: The purpose of our research is to understand the subjective well-being (SWB) of Chinese adult residents and its influencing factors and to identify the key groups and areas to provide a basis for the formulation of relevant policies to improve residents’ happiness. (2) Methods: In this study, we analyzed the influencing factors of SWB of individuals older than 16 years of age, according to the 2014 China Family Panel Study (CFPS). We weighted 27,706 samples in the database to achieve the purpose of representing the whole country. Finally, descriptive statistics were used for the population distribution, chi-square tests were used for univariable analysis, and binary logistic models were used for multivariable analysis. (3) Results: The response rate of SWB was 74.58%. Of the respondents, 71.2% had high SWB (7–10), with a U-shaped distribution between age and SWB. Females are more likely than males to rate themselves as happy. There is a positive ratio between years of education and SWB. Residents who have better self-evaluated income, self-rated health (SRH), psychological well-being (PWB), Body Mass Index (BMI), social trust, social relationships, and physical exercise have higher SWB. (4) Conclusion: The results of the present study indicate that to improve residents’ SWB, we should focus more attention on middle-aged and low-income groups, particularly men in agriculture. The promotion of SWB should be facilitated by improvements in residents’ education, health status, and social support as well as by the promotion of smoking bans and physical exercise.

## 1. Introduction

Research on residents′ subjective well-being (SWB) can promote the improvement of relevant policies and related fields. Researchers and policymakers have realized the importance of SWB as a solid indicator of the economic and social well-being of societies as a whole [1]. According to Diener and Suh, “SWB research is concerned with individuals’ subjective experiences of their lives. The underlying assumption is that well-being can be defined by people’s conscious experiences—in terms of hedonic feelings or cognitive satisfactions. The field is built on the presumption that to understand the individuals’ experiential quality of well-being, it is appropriate to directly examine how a person feels about life in the context of his or her own standards” [2]. The multidimensional concept of SWB is associated with a diversified range of correlates (i.e., demographics, education, income, health, psychosocial resources) [3]. SWB could be measured by the assessment of how frequently or intensely people experience a variety of positive and negative emotions, such as “happiness,” “sadness,” “anger,” or “joyfulness” [4].

There is evidence that health and SWB are closely related [5]. Poor health leads to increased mortality rather than joy [6]. Increasing evidence shows that psychological well-being (PWB) may be a protective factor in health, reducing the risk of chronic disease and promoting longevity [7]. With the improvement in social and economic conditions, obesity has become a burden for an increasing number of residents; thus, scholars have focused on the impact of BMI on SWB, and their conclusions are not the same [8,9]. Therefore, our article explores the specific relationship between BMI and residents’ SWB.

Generalized trust is a valuable social resource, not only for the individual but also for society as a whole [10]. Social trust is increasingly considered a psychosocial determinant of SWB, and an SWB survey conducted among Iranian women in developing countries, such as China, showed that trust is positively correlated with better SWB [11].

Evidence has long been accumulating concerning the association between social relationships and health and well-being at all ages [12]. There is convincing evidence that poor social relationships negatively impact PWB [13]. To date, systematic reviews have summarized the links between social relationships and PWB in able-bodied populations [14]. Among life’s most intimate relationships, marriage has been found to be positively associated with better health for several reasons. Marriage as a social relationship affects PWB in two alternative (although not mutually exclusive) causal models, namely, the main-effect model and the stress-buffering model. The stress-buffering model posits that social ties are related to well-being only for persons under stress, whereas the main-effect model proposes that social relationships have a beneficial effect regardless of whether individuals are under stress [13]. However, it has also been concluded that unsatisfactory marital or family relationships of many depressed older adults are often consequences rather than predictors of PWB [15]. Although marital dissatisfaction with partners is uncommon, it is highly correlated with depressive symptoms [16], which undoubtedly indirectly affects SWB.

Several studies have established a positive relationship between levels of habitual physical activity (PA) and SWB [17,18]. Exercise indirectly affects the quality of life of residents by affecting their health [19], and in particular, it can have a significant impact on the SWB of working-age elderly people [20]. More direct evidence suggests that participation in physical exercise is almost always associated with better SWB, especially among participants with low or high levels of physical exercise rather than intermediate levels [21]. Some studies examining the relationship between PA intensity and SWB in healthy adults have found different results, one of which is that PA has no effect on SWB [22,23].

The relationship between smoking and SWB remains controversial in existing studies. The famous Alameda Seven study in 1965 in Alameda County, California, found that smoking affects health, and poor health may reduce SWB levels [24]. However, some smokers believe that quitting smoking reduces their overall SWB and impairs their ability to socialize and cope with stress [25].

Existing studies show a complex relationship between income and SWB. The “Easterlin paradox” notes that the growth of money does not necessarily lead to the growth of SWB; as material wealth accumulates, it cannot promote the increase in SWB after a certain stage [26]. In this regard, social comparison theory divides people into two groups: those who tend to compare themselves with those who do better than themselves, and those who tend to compare themselves with those who do worse than themselves. Moreover, most individuals tend to compare upward [27]. Thus, this psychological trend partly explains the increase in people’s income but not necessarily the year-after-year increase in SWB.

As a measure of quality of life, SWB has drawn considerable attention. However, there are still some deficiencies in the existing research on SWB in China. On the one hand, it is limited to the exploration of SWB by a few specific influencing factors, and on the other hand, it is limited by conditions. Therefore, it is difficult to consider the research as including the entire country. In our study, we aim to discuss the impact of sociodemographic characteristics, physical health status, PWB status, social trust, social relations, physical exercise, and smoking on SWB among a nationally representative sample.

## 2. Materials and Methods

### 2.1. Data Sources and Sample Composition

The data are from China Family Panel Studies which was funded by “985 Program” of Peking University and carried out by the Institute of Social Science Survey of Peking University. CFPS is a nationally representative, annual longitudinal survey administered by the Institute of Social Science Survey (ISSS) of Peking University. The CFPS survey was reviewed and approved by the ISSS of Peking University. All participants were asked to provide written informed consent. The data were released to the researchers without access to any personal data and included responses from 37,147 Chinese individuals residing in 621 villages/communities from 25 of China’s 30 provinces. All the subsampling frames of CFPS were obtained through a stratified three-stage (districts/counties-villages/communities-households) probability random sampling procedure. The primary sampling unit (PSU) was administrative districts (counties). The second-stage sampling unit (SSU) was administrative villages (communities). The third-stage (ultimate) sampling unit (TSU) was households. Within each household, members aged 16 years and older were selected as the respondents [28]. During all stages of data collection, the research team adopted a telephone check, field check, audio record check, interview reviews, and statistical analyses to ensure data quality. The survey questionnaire contained detailed individual information regarding physical health status, PWB, social trust, social relations, physical exercise, smoking status, and self-evaluated income status, which made the CFPS the ideal dataset for our study on SWB. The subjects in our study were 16 years of age or older. Approximately 88.48% of the residents answered the question regarding SWB. After missing values for some variables in the dataset were considered, our final analytical sample consisted of 27,706 respondents. The overall response rate was 74.58%.

### 2.2. Variables and Definitions

#### 2.2.1. Dependent Variable

The dependent variable was SWB, which was based on the resident’s response to “What is your happiness level?” Happiness in the questionnaire in Chinese translation contains the sum of the resident’s long-term positive emotions and negative emotions, which comprises SWB. The answers were recorded from 0 (lowest score) to 10 (highest score), with 0–6 denoting low SWB and 7–10 denoting high SWB, referring to the classification of the score (0–10) by the residents on the questionnaire (very low, low, medium, high, very high).

#### 2.2.2. Independent Variables

We recorded a variety of sociodemographic variables as follows: age, gender, area of residence, years of education, marital status, type of work, and self-evaluated income status. Age in years was coded into the following six categories: 0: ≤24; 1: ~34; 2: ~44; 3: ~54; 4: ~64; and 5: ≥65. Gender was dichotomously coded as male (1) and female (0). Area of residence was categorized into urban (1) and rural (0). Years of education were coded into four categories based on the number of completed years in the Chinese education system as follows: 1: ≤6; 2: ~9; 3: ~12; and 4: ≥13. Marital status was coded into the following three categories: never married (1); married or cohabitating (2); and widowed or divorced (3). Type of work was categorized into the following three types: inapplicable (0); agriculture (1); and non-agricultural (2). The status of residents’ self-assessed income was recorded via answers to the item “Your relative income level in the local area”, with possible responses ranging from 1 (very low) to 5 (very high). The answers were coded into the following four categories: low (1); general (2); high (3); and inapplicable (0).

Health status was evaluated according to two aspects of physical and psychological well-being (PWB). Previous studies have consistently shown that self-rated health (SRH) is a valid and reliable indicator of morbidity and mortality [29]. In our study, the SRH was recorded on a five-point scale as follows: 1 = excellent; 2 = very good; 3 = good; 4 = not good; and 5 = poor. In addition, we used the following three indicators that are commonly used in health services research to further evaluate physical health: two-week morbidity rate (whether participants felt physical discomfort in the past two weeks), morbidity of chronic disease (whether participants had experienced a doctor-diagnosed chronic disease in the past six months) and hospitalization rate (whether participants were hospitalized due to illness in the past 12 months).

We used the K6 scale developed by Kessler et al. to indicate PWB [30]. The K6 scale has been widely shown to be an effective measure of PWB [31,32]. The six questions of the K6 scale are as follows: (1) How often do you have trouble getting excited? (2) How often do you feel nervous? (3) How often do you fidget and lose your cool? (4) How often do you despair about the future? (5) How often do you find it hard to do anything? (6) How often do you find your life meaningless? The answers (always, almost every day, half the time, sometimes and never) were scored as 4, 3, 2, 1 and 0, respectively. The total score of the K6 scale is the sum of the scores of the six questions; thus, the total score ranged from 0 to 24. A higher score indicates a worse PWB status. In this study, the total PWB score was subdivided into the following two categories according to Kessler’s division: low risk for mental disorders (0–12) and high risk for mental disorders (13–24).

Body shape can affect a person’s physical and PWB. Therefore, we also explored the relationship between BMI and SWB. BMI was coded into the following four categories according to the guidelines on the prevention and treatment of overweight and obesity among Chinese adults: 0: < 18.5 (underweight); 1: 18.5–23.9 (normal); 2: 24–27.9 (overweight); and 3: ≥28 (obesity) [33,34].

Our study recorded social trust and social relations. Social trust was assessed by asking residents the following question: “In general, do you agree that most people are trustworthy?” We coded trustworthy as 1 if the answer was “yes, most are trustworthy” and untrustworthy as 0 if the answer was “we should be as careful as possible”. Social relationship was measured by asking respondents the following question: “How would you rate your ability to relate to others?” To assess social relations, we condensed the total score for social relationships (maximum of 10 points) into a binary variable according to the classification of a 0 to 10 score on the questionnaire by the residents: 0 = 1−6 and 1 = 7−10. At the same time, marital satisfaction was documented as a type of social relation, with scores ranging from 1 (very satisfied) to 5 (very dissatisfied). We grouped these into inapplicable (0), satisfaction (1), general (2), and dissatisfaction (3).

Physical exercise was determined by asking about the “residents’ physical exercise frequency in the last month when not on vacation”. We grouped these responses as never (0 times), sometimes (1–4 times) and often (≥4 times). Smoking status was measured by asking whether a respondent was currently smoking (1 =  yes and 0 = no).

### 2.3. Statistical Analysis

IBM SPSS Statistics version 24 (IBM Corp, Armonk, NY, USA) was used for data processing. Given the sampling weights specified in the CFPS design, this study weighted the data to match the composition of the Chinese population in 25 provinces and provincial cities [35]. Descriptive statistics were computed for the sample, chi-square tests were performed for univariable analysis, and binary logistic models were used for multivariable analysis (significance level set at *p* < 0.05).

## 3. Results

Table 1 shows the frequency and distribution of the characteristics of the number of participants in the actual sample and weighted processing. Among the 27,706 residents, the majority were 35~44 years of age (20.6%) and 45~54 years of age (20.9%). Approximately 68.4% of people had nine years of education or less. The self-evaluated income status of most residents was low (39.5%) and general (41.7%). Approximately 72.2% of people rated their health as good or better. Furthermore, according to the results of the K6 scale, 95.2% of them had high levels of PWB, and the proportion of residents with a normal BMI range was 58.1%.

Table 2 displays the distribution of high SWB among people with different characteristics. After weighted treatment, sociodemographic characteristics, physical and psychological health status, social relations, social trust, smoking, and physical exercise all had an impact on residents’ SWB (*p* < 0.05). The residents with high SWB accounted for 71.2%. The age group with the highest SWB was 16 to 24 years (79.5%), and the age group with the lowest SWB was 45 to 54 years (66.5%). Residents with more than 13 years of education reported the highest percentage of high SWB (80.9%). Residents with better physical and psychological health status had high SWB, and those with higher BMI had higher SWB. Residents with high social trust and social relationships had higher SWB (76.2%, 83.4%). Moreover, the more physical exercise that the residents performed, the higher the proportion of residents with high SWB.

Table 3 shows the odds ratios of SWB by different resident characteristics. The SWB of middle-aged people was lower than that of young people and older people, as shown by the U-shape. Women were more likely than men to rate their SWB as high (OR = 1.378; 95% CI = 1.245–1.525). People who live in rural areas were less happy than those who live in cities (OR = 0.865; 95% CI = 0.793–0.945). People who work in agriculture were less happy (OR = 0.810; 95% CI = 0.728–0.901). The table shows that years of education, self-evaluated income status, SRH, BMI, marital satisfaction, and physical exercise were significantly related to increased odds of high SWB with an approximately positive correlation. Residents with poor PWB were less happy than those with positive PWB (OR = 0.425; 95% CI = 0.535–0.512). The effect of social trust (OR = 0.748; 95% CI = 0.689–0.811) and social relationships (OR = 0.181; 95% CI = 0.166–0.196) on SWB was significant; the better the resident’s social trust and social relations were, the more likely they were to have high SWB. In addition, the odds of SWB among nonsmoking residents were 1.179 times the odds of SWB among smoking residents (95% CI = 1.056–1.316).

## 4. Discussion

After weighting, our study showed that approximately 71.2% of the residents of China have high SWB, which is basically consistent with our expectation. Since the 1980s, great changes have taken place in Chinese society. Although on the one hand, social problems, such as the widening gap between the rich and the poor, the increasing inequality in education, and the huge impact on traditional marriage and family values, have emerged, there is no denying that China’s economic growth, educational expansion, demographic transformation, and international status have brought huge practical benefits to Chinese residents. Therefore, it is not surprising that residents have high SWB, but future changes still warrant our attention.

### 4.1. Sociodemographic Characteristics and SWB

The results of this study showed that the distribution of higher SWB in the age group presented a U-shaped curve, similar to that found in several European, American, Asian, and Latin American cross-sectional surveys over several time periods. Furthermore, these results replicated prior findings of a U-shaped association between age and well-being, with the nadir at middle age and higher well-being among younger and older adults [36]. In addition, gender and marital status have been shown to have consequences for the SWB of residents. Women have higher SWB levels than men, and residents who were married or cohabitated had higher SWB. Females are more likely to engage in social activities and the performing arts, whereas males are more likely to engage in detachment-recovery and aesthetic activities [37,38]. The reason may be that with the improvement in China’s economy and civilization, the social status of Chinese females has changed, along with an increase in social activities and an increased probability of eliminating adverse lifestyle factors. It is possible that the social support offered by marriage exerts a protective effect for some men [39]; to some extent, this conclusion can explain why residents who are divorced or widowed have the lowest SWB in this study. Our results showed that people’s SWB increases as their education level increases. The effect of years of education may come from two aspects. First, education can change individuals’ cognitive ability to understand people and things, influence their ability to obtain stable emotional support, and increase their SWB. Second, education affects people’s work, income, and social status, thus affecting people’s SWB [40]. Our results showed that most people (68.4%) have only 9 years of education or less; therefore, the overall cultural quality of residents should be improved. It is worth noting that there is no difference in the happiness level of people with primary school education (6 years) and junior high school education (9 years), while the subjective happiness of people with more than 10 years of education increases, suggesting that we should strengthen higher education on the basis of universal nine-year compulsory education and provide opportunities for people to receive higher education.

This study found that people who live in rural areas and engage in agricultural work have lower SWB. The reason may be that living in urban areas and having non-agricultural jobs can enable residents to improve their objective material conditions (economic income, social class) based on human capital theory and status acquisition theory [41]. Our discussion of the relationship between income levels and SWB also supports this view. The Easterlin paradox pointed out that economic income was positively correlated with happiness; however, when economic income increased to a certain level, SWB decreased instead [42]. In terms of self-evaluated income, we concluded that self-evaluated income is positively correlated with SWB, which is not indicated by the Easterlin paradox, which reminds us that although the country is developing and making progress, the overall economic level of the country needs to be improved.

### 4.2. Health Status and SWB

SRH is widely accepted as a means of reporting physical health and psychological health and serves as an indicator of morbidity and mortality [43]. Our study showed that 72.2% of residents think that their health was good or better, and residents’ SRH was positively correlated with SWB. SWB was also correlated with hospitalization rates. The two-week morbidity rate and morbidity of chronic disease were meaningful in the single-factor analysis and meaningless in consideration of the influence of other factors in the multifactor analysis. Surprisingly, in our study, the results based on the resident hospitalization rate were contrary to our expectations; residents who answered “yes” were happier than those who answered “no” (OR = 0.860; 95% CI = 0.749–0.987). The reason may be that a small number of residents (10.4%) have lower expectations for their standard of living after illness and discharge, and therefore, they are more likely to have higher SWB. All of the above results suggest that physical health is closely related to SWB, and we should strengthen the development of health care and meet peoples basic needs for health services.

At the same time, it is worth noting that the BMI of residents was positively correlated with SWB, which differs from our expectation that SWB is highest among people with normal BMI. This result may be related to the differences in the definition of obesity and cultural differences between China and the West. The World Health Organization (WHO) defines obesity as a BMI >30 according to the BMI standard of the European population and subdivides it into three levels. With the increase in BMI, individuals’ ability to perform normal life functions will be increasingly affected. However, the obesity levels of Chinese residents may not be sufficient to affect their normal life functions. In addition, a Chinese cultural saying of “Laugh and grow fat” also indicates that weight is closely related to SWB, and the causal relationship between the two needs to be further explored.

As for PWB, according to the K6 scale, residents with different risks of mental illness have significant differences in SWB, and residents with poor PWB are far less likely to have high SWB than those with high PWB. On the one hand, positive psychological health has beneficial effects on health and the survival of the population [44,45]. On the other hand, the broaden-and-build theory of positive emotions notes that positive emotions are the method of adaptability in human evolution, and they can facilitate the building of lasting resources to achieve better results, including financial support, social relations, and even higher levels of resilience [46,47]. This idea suggests that the mental health of our residents is very important to SWB and that mental health services need to be strengthened.

### 4.3. Social Support and SWB

Our study confirmed earlier findings that social trust and social relationships were positively and significantly associated with SWB [14]. On the one hand, social relationships may affect SWB through health because social relationships have been shown to influence SRH through lifestyle [48]. One possible explanation is that in China, people are more collectively oriented. Social relationships (especially with intimate persons) form an important part of daily life and are frequently centered around food and drink. During the process of interaction, social relationships can affect health behavior through peer effects (e.g., role models). These behaviors can be positive or negative, depending on the lifestyle of the intimate person. On the other hand, social relationships may provide different kinds of social support; the literature on social support further distinguishes between emotional support (e.g., someone being available to listen or offer sympathy during times of crisis or hardship, or someone available to give advice) and instrumental support (e.g., someone available to help with issues that require physical effort or financial aid). All these different forms of social support appear to have different implications for mental health. Support may also be provided to or received from different sources, such as spouses, children, relatives, friends, and coworkers to some extent, which can explain why people with positive social relations have higher SWB. The results of our study showed that residents with higher marital satisfaction had higher SWB levels, which may be because a benign marriage, similar to a social relationship, can be transformed into a kind of social support and increase residents’ SWB. As we mentioned above, married people reported higher SWB than those who were unmarried and widowed or divorced, which also provides evidence of marriage (the most intimate social relationship) as an influencing factor of SWB.

Social trust as a form of social capital was positively correlated with SWB, and trust affects health and SWB through social networks and support [49], which is largely consistent with our conclusion. Some studies explain the relationship between social trust and SWB from the perspective of intelligence, and intelligent individuals may be better at identifying when any particular person is likely to act in an untrustworthy manner based on the characteristics of the prospective interaction (e.g., material payoffs, discount rates). Alternatively, it may simply be that intelligent individuals have a greater chance of interacting with people who are materially better off and who therefore have less to gain from acting in an untrustworthy way [10]. From the perspective of spiritual culture, social trust may be closely related to the sense of survival, sense of security, sense of dignity, and sense of happiness, all of which may influence each other. Thus, the causal relationship between social trust and SWB is still worthy of exploration.

### 4.4. Physical Exercise, Smoking, and SWB

Our results showed that nonsmokers were more likely to report high SWB. Some research has shown that compared with nonsmokers, current smokers were more likely to report poor general health [50], which may partly explain the difference in SWB between residents who smoke and those who do not. Another reason is that China’s powerful propaganda campaign to promote awareness of the harmful effects of smoking has helped to prevent the spread of smoking via social interaction. At the same time, our studies showed that smokers have lower SWB levels, confirming what some studies have shown, namely, that residents with low SWB levels may use smoking to relieve stress and for social interaction [25]. For physical exercise, our research showed that most residents never exercised purposefully for a month (61.9%); meanwhile, residents who exercised 1 to 4 times a month accounted for 16.8% of the sample, and those who exercised more than 4 times accounted for 21.3% of the sample. With the increase in exercise frequency, SWB also increased. There is evidence that levels of physical exercise can improve the health of residents, thereby increasing SWB [51]. Another explanation might be that people who exercise purposefully have a positive attitude toward life and have plenty of time and energy; therefore, they have higher SWB. This conclusion shows that exercise awareness should be publicized nationwide to improve exercise levels.

## 5. Strengths and Limitations

### 5.1. Strengths

Our research has some methodological advantages. First, we were able to make nationwide extrapolations from a high-probability sample collected from across the country through the use of a weighted analysis rather than a single sample. Thus, greater heterogeneity of the population was captured in this study. Second, we included relatively comprehensive influencing factors for analysis according to existing research. Third, the tests of validity and reliability in this population were performed by an investigation team, which guaranteed the quality of the investigation and provided a solid foundation for our study.

### 5.2. Limitations

Most of our factor measurements were evaluated by single-question, self-reported items on a questionnaire. As a result, they may suffer from recall bias and misclassifications. The feedback effect between the mediator and the dependent variable can cause simultaneity bias. Because this study is the conclusion of observational research, further confirmation is needed.

## 6. Conclusions

We analyzed the influence of various factors on SWB in this cross-sectional research using data from a large Chinese panel (CFPS). We found that many factors may have a major impact on SWB, which provides strong policy support for our policymakers, thus facilitating a focus on key populations and groups. We hope that the results will help health professionals find the key population and provide evidence for policymakers to improve relevant policies. We may focus more on the SWB of middle-aged people and low-income groups, particularly men in agriculture, and the promotion of SWB may be facilitated via the improvement of residents’ education, health status, social support, and the promotion of smoking bans and physical exercise.

## Figures and Tables

**Table 1 ijerph-16-02566-t001:** Characteristics of the study sample.

Characteristic		*N*	Weighted Composition Ratio [%(SE)]
Age in years	16–24	3211	14.3 (0.3)
	~34	4173	17.7 (0.3)
	~44	5077	20.6 (0.3)
	~54	6089	20.9 (0.3)
	~64	5128	15.3 (0.3)
	≥65	4028	11.2 (0.2)
Gender	Female	13,757	50.0 (0.4)
	Male	13,949	50.0 (0.4)
Area of residence	Rural	14,302	39.6 (0.4)
	Urban	13,404	60.4 (0.4)
Years of education	0–6	11,168	35.0 (0.4)
	7–9	9208	33.4 (0.4)
	10–12	4568	18.7 (0.3)
	≥13	2762	12.8 (0.3)
Type of work	Agricultural	10,182	30.9 (0.4)
	Non-agricultural	10,767	44.8 (0.4)
	Inapplicable	6757	24.4 (0.4)
Self-evaluated income status	Low	10,971	39.5 (0.4)
	General	11,693	41.7 (0.4)
	High	2775	9.1 (0.2)
	Inapplicable	2267	9.8 (0.3)
Marital status	Never married	3760	17.1 (0.3)
	Married/cohabitation	22,030	76.7 (0.4)
	Divorced/widowed	1916	6.2 (0.2)
SRH	Excellent	3894	14.9 (0.3)
	Very good	5762	21.9 (0.3)
	Good	9774	35.5 (0.4)
	Not good	3999	14.4 (0.3)
	Poor	4277	13.4 (0.3)
Two-week morbidity rate	No Yes	19,444 8262	71.7 (0.4) 28.3 (0.4)
Morbidity of chronic disease	No Yes	23,023 4683	85.1 (0.3) 14.9 (0.3)
Hospitalization rate	No Yes	24,680 3026	89.6 (0.2) 10.4 (0.2)
BMI	Underweight	2610	9.3 (0.2)
	Normal	16,007	58.1 (0.4)
	Overweight	7161	25.3 (0.3)
	Obesity	1928	7.2 (0.2)
PWB	13–24	1443	4.8 (0.2)
	0–12	26,263	95.2 (0.2)
Social trust	Be as careful as possible	12,763	46.0 (0.4)
	Most are trustworthy	14,943	54.0 (0.4)
Social relationship	0–6	8781	30.8 (0.4)
	7–10	18,925	69.2 (0.4)
Marital satisfaction	Dissatisfied	870	2.9 (0.1)
	General	2034	7.2 (0.2)
	Satisfied	19,126	66.5 (0.4)
	Inapplicable	5676	23.3 (0.4)
Smoking	No	19,672	72.7 (0.4)
	Yes	8034	27.3 (0.4)
Physical exercise	Never	17,403	61.9 (0.4)
	Sometimes	4195	16.8 (0.3)
	Often	6108	21.3 (0.3)
Total		27,706	100

Note: SRH, self-rated health; PWB, psychological well-being.

**Table 2 ijerph-16-02566-t002:** The distribution of high subjective well-being (SWB) and univariable analysis.

Variables		High SWB [%(SE)]	X^2^	*p*
Age in years	16–24	79.6 (0.9)	251.268	<0.001
	~34	74.3 (0.9)		
	~44	68.8 (0.8)		
	~54	66.5 (0.8)		
	~64	68.8 (0.9)		
	≥65	72.1 (1.0)		
Gender	Female	72.7 (0.5)	31.328	<0.001
	Male	69.7 (0.5)		
Area of residence	Rural	67.4 (0.5)	126.354	<0.001
	Urban	73.7 (0.5)		
Years of education	0–6	64.7 (0.6)	462.126	<0.001
	7–9	70.8 (0.6)		
	10–12	77.4 (0.8)		
	≥13	80.9 (1.0)		
Type of work	Agricultural	64.1 (0.6)	307.434	<0.001
	Non-agricultural	74.1 (0.6)		
	Inapplicable	75.0 (0.7)		
Self-evaluated income status	Low	63.0 (0.6)	629.747	<0.001
	General	75.3 (0.5)		
	High	78.3 (1.1)		
	Inapplicable	80.3 (1.1)		
Marital status	Never married	74.9 (0.9)	169.290	<0.001
	Married/cohabitation	74.1 (0.4)		
	Divorced or widowed	58.4 (1.5)		
SRH	Excellent	82.8 (0.8)	1016.035	<0.001
	Very good	78.2 (0.7)		
	Good	71.8 (0.6)		
	Not good	61.2 (1.0)		
	Poor	56.2 (1.0)		
Two-week morbidity rate	No	73.9 (0.4)	251.202	<0.001
	Yes	64.3 (0.7)		
Morbidity of chronic disease	No	71.9 (0.4)	40.742	<0.001
	Yes	67.1 (0.7)		
Hospitalization rate	No	71.5 (0.4)	7.552	<0.05
	Yes	69.0 (1.1)		
BMI	Underweight	66.6 (1.2)	48.273	<0.001
	Normal	70.8 (0.5)		
	Overweight	72.7 (0.7)		
	Obesity	74.9 (1.3)		
PWB	13–24	40.2 (1.8)	656.806	<0.001
	0–12	72.8 (0.4)		
Social trust	Be as careful as possible	65.4 (0.6)	391.333	<0.001
	Most are trustworthy	76.2 (0.5)		
Social relationship	0–6	43.7 (0.7)	4539.591	<0.001
	7–10	83.4 (0.4)		
Marital satisfaction	Dissatisfied	36.2 (2.2)	1893.205	<0.001
	general	37.2 (1.4)		
	Satisfied	76.7 (0.4)		
	Inapplicable	70.5 (0.8)		
Smoking	No	72.8 (0.4)	87.747	<0.001
	Yes	67.1 (0.7)		
Physical exercise	Never	67.4 (0.5)	318.798	<0.001
	Sometimes	76.8 (0.9)		
	Often	77.8 (0.7)		
Total		71.2		

Note: Weighted data are used. SWB, subjective well-being; SRH, self-rated health; PWB, psychological well-being.

**Table 3 ijerph-16-02566-t003:** Odds ratios with 95% confidence intervals of the association between SWB and various factors.

Variables		OR	Lower 95% CI	Upper 95% CI
Age in years	16–24	1.000		
	~34	0.635	0.513	0.787
	~44	0.518	0.410	0.654
	~54	0.498	0.394	0.629
	~64	0.678	0.532	0.864
	≥65	0.917	0.708	1.189
Gender	Male	1.000		
	Female	1.378	1.245	1.525
Area of residence	Urban	1.000		
	Rural	0.865	0.793	0.945
Years of education	0–6	1.000		
	7–9	1.069	0.964	1.185
	10–12	1.217	1.061	1.397
	≥13	1.273	1.063	1.525
Type of work	Non-agricultural	1.000		
	Agricultural	0.810	0.728	0.901
	Inapplicable	1.059	0.928	1.208
Marital status	Married/cohabitation	1.000		
	Never married	0.449	0.373	0.542
	Divorced/widowed	0.010	0.340	0.473
Self-evaluated income status	Low	1.000		
	General	1.448	1.324	1.584
	High	1.707	1.470	1.983
	Inapplicable	1.640	1.361	1.977
SRH	Poor	1.000		
	Not good	1.133	0.978	1.313
	Good	1.553	1.353	1.783
	Very good	2.008	1.713	2.354
	Excellent	2.793	2.336	3.341
Two-week morbidity rate	Yes	1.000		
	No	1.073	0.972	1.183
Morbidity of chronic disease	Yes	1.000		
	No	0.924	0.818	1.045
Hospitalization rate	Yes	1.000		
	No	0.860	0.749	0.987
BMI	Underweight	1.000		
	Normal	1.184	1.027	1.365
	Overweight	1.262	1.078	1.477
	Obesity	1.521	1.237	1.869
PWB	0–12	1.000		
	13–24	0.425	0.353	0.512
Social trust	Most are trustworthy	1.000		
	Be as careful as possible	0.748	0.689	0.811
Social relationship	7–10	1.000		
	0–6	0.181	0.166	0.196
Marital satisfaction	Dissatisfied	1.000		
	General	1.031	0.802	1.325
	Satisfied	4.469	3.600	5.547
	Inapplicable	4.469	3.600	5.547
Smoking	Yes	1.000		
	No	1.179	1.056	1.316
Physical exercise	Never	1.000		
	Sometimes	1.126	0.996	1.273
	Often	1.311	1.175	1.463

Note: Weighted data are used. SWB, subjective well-being; SRH, self-rated health; PWB, psychological well-being.

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
