# Peer review of "Analysis of Factors Affecting the High Subjective Well-Being of Chinese Residents Based on the 2014 China Family Panel Study"

_ijerph, 2019, doi:10.3390/ijerph16142566_

Round 1

Reviewer 1 Report

Dear authors,

The article I have now reviewed is highly interesting and posits itself within the continuum of a relativley new socio-medical  research stream. 

The overall structure and content was good and needs no futher actions. However, I would like to point some parts of the article to be considered for reformulation.

1. This sentence: We hope the results will help health professionals find the key population and provide evidence for 96 policy makers to improve relevant policies. would better serve in Conclusions, not in the introduction.

2. The literature about subjective well-being is vast, as are the indicators reflecting it. It is common to use simple one-item variables to measure SWB, like is done in this article. However, I encourage the authors to problematise little more the one-item-variable's ability to measure SWB, in this case the level of happiness. Is it sufficient enough to capture the content of SWB? Because SWB is measured by happiness alone, I suggest relabelling the outcome variable (dependent) as happiness and reformulate the topic of the paper accordingly.

3. Overall, the section concerning variables and definitions is well-formed. However, I ask the authors to clarify shortly the methodological choices related to the setting of cut-points. That is, based on what low SWB is set to be between 0-6? This concerns also the other continous variables used. 

Otherwise this article seems erady for publication.

Author Response

We appreciate your suggestion very much, and we have made corresponding modifications.

Reviewer 2 Report

The study aims at identifying factors contributing to difference in subjective well-being in Chinese residents. Although the sample is impressive and a wide range of potential factors is considered, the paper lacks a profound research question and the analysis of the data is inadequate. The description of the data collection, testing of data quality and methods is incomplete. I would suggest using the SWB variable with its interval scale properties rather than dichotomizeing it. Further, a profound discussion of the results in terms of the measurement of SWB is needed. Since the results are cross-sectional and on a descriptive level, the conclusions are not justified.

Author Response

We appreciate your suggestion very much, and we have made corresponding modifications.Please see the attachment.

Reviewer 3 Report

Thank you for inviting me to review the study "Analysis of factors affecting the subjective well-2 being of Chinese residents based on the 2014 China 3 Family Panel Study (CFPS)". The study provided interested results and information for research and health policies.

Comments:

1.      With respect to the research ethics, approval from an ethics committee should have been obtained before undertaking the research. At a minimum, a statement including the date of approval and name of the ethics committee or institutional review board should be cited in the “Materials and Methods” section of the article.

2.      With respect to the dependent variable, on what evidence have they based to recoded “0-6 denoting low SWB and 7-10 denoting high SWB”?

3.      Given that the dependent variable in the logistic regression is "high SWB", I propose that "high SWB" be included in the title of the manuscript.

Author Response

(The authors gave the same response as above.)

Round 2

Reviewer 3 Report

The authors responded to my comments.

Author Response

Thank you